# I Don’t Want to Be Thin! Fear of Weight Change Is Not Just a Fear of Obesity: Research on the Body Mass Anxiety Scale

**DOI:** 10.3390/ijerph20042888

**Published:** 2023-02-07

**Authors:** Wojciech Styk, Ewa Wojtowicz, Szymon Zmorzynski

**Affiliations:** 1Department of Psychology, Medical University of Lublin, 20-059 Lublin, Poland; 2Polish Academy of Social Sciences and Humanities, 69 Banstead Road, Carshalton, London SM5 3NP, UK; 3Department of Cancer Genetics with Cytogenetic Laboratory, Medical University of Lublin, 20-059 Lublin, Poland

**Keywords:** body mass anxiety scale, body mass index (BMI), obesity, weight change

## Abstract

Anxiety is one of the psychological factors associated with body weight experienced by people attempting to live up to expectations of an ideal body shape. The stigma of excessive or too low body weight and the stigmatization of people because of it is becoming a widespread problem with negative psychological and social consequences. One effect of the strong social pressure of beauty standards dependent on low body weight is the development of eating disorders and negative societal attitudes toward overweight or obese people. Research conducted to date has mainly focused on one dimension of weight-related anxiety—the fear of getting fat. Ongoing research has also revealed the other side of weight-related anxiety—fear of weight loss. Therefore, the purpose of the present project was to develop a two-dimensional scale to diagnose the level of weight-related anxiety and to preliminarily test the psychometric properties of the emerging constructs. Results: the BMAS-20 weight-related anxiety scale in both Polish and English versions was developed and its psychometric properties were confirmed. The components of body weight-change anxiety that emerged were: anxiety about getting fat and anxiety about losing weight. It was found that both AGF and ALW may have a protective function related to awareness of the negative consequences of poor eating and the health risks associated with it. Above-normal levels of anxiety may be a predictor of psychopathology. Both AGF and ALW are associated with symptoms of depression.

## 1. Introduction

Diets have become a dominant part of social discourse around the world. They can be considered part of a healthy lifestyle in different cultures.. Eating behaviors are complex and underpinned by biological and environmental factors [1,2,3]. Appropriate body-weight awareness is associated with two motivational orientations with different directions: the anxiety about becoming overweight or the desire to achieve a slim figure [4]. Self-control in food intake can manifest itself as restrictions in food intake and inappropriate daily diet, which clearly have an impact on body weight. 

The stigma of being overweight or underweight is becoming a widespread problem with negative psychological and social consequences. The effects of such stigma result in strong social pressure to conform to societal beauty standards dependent on low body weight. Moreover, such pressure can lead to the development of eating disorders and socially negative attitudes towards overweight or obese people. The internalization of negative attitudes about body shape also contributes to one’s self-evaluation [5]. Current research primarily focuses on finding links between overweight and obesity and psychological factors promoting body mass maintenance. However, new weight-related phenomena are currently appearing in relation to people being perceived as too thin [6]. Health effects are also emphasized, highlighting the dangers of eating disorders. The causes of eating disorders are complex and multifactorial, with co-occurring cultural, psychological, and/or family and social factors playing a role in their activation [7,8]. Although genetic factors have also been identified, environmental and social factors are considered to be crucial [9].

One of the psychological factors associated with body weight is the anxiety experienced by people trying to live up to the expectations of an ideal figure. Anxiety is defined as a non-specific, unpleasant emotional state characterized by experiencing worry, fear, stress and annoyance [10]. Anxiety is often contrasted with fear due to the fact that anxiety is a state without an object, whereas fear is always a fear from someone, something or some event. It is well known that anxiety disorders co-occur with mood disorders and depression [11].

Anxiety about weight gain can be defined as a reaction in response to stimuli whose evaluation may be negatively associated with increasing body weight. As a result, maladaptive behaviors may emerge, manifesting as eating restrictions and dietary regimes, as consequences of weight gain anxiety or avoidance of weight gain [12,13].

In the case of excessive weight anxiety, dietary behavior refers to the restriction of food intake, usually in order to control weight [14]. Anxiety about being overweight is linked to dissatisfaction with one’s own body and contributes to the development of eating disorders [15]. The anxiety co-occurs with obesity [16], emotional eating and paroxysmal eating [17,18,19], and anorexia nervosa or bulimia [20,21]. Thus, anxiety disorders may favor engaging in a restrictive approach to dieting. In contrast, analyses of weight-loss anxiety, as well as the modification of eating behaviors and diets to increase caloric intake are rare in the literature. The rapid growth of the obesity epidemic justifies the asymmetric interest of researchers in addressing its associated phenomena.

Research to this date has mainly focused on one dimension of weight-related anxiety —anxiety about getting fat. Commonly used in research, The Goldfarb Fear of Fat Scale is a 10-item scale developed in the 1980s [22]. Studies indicate different structures depending on the clinical group—single or two-factor. In non-clinical groups, a two-factor structure consisting of anxiety about gaining weight and anxiety about losing control over food/weight is indicated [23].

Weight-related disorders and the whole spectrum of eating disorders are more or less confronted with anxiety from different origins. As mentioned above, the clinical criteria of anorexia, for example, include an excessive desire for weight loss, which may be accompanied by inappropriate self-esteem, distorted body perceptions and a morbid anxiety about gaining weight. In contrast, our own experience in the clinical work with patients struggling with overweight and obesity indicates a characteristic anxiety occurring, among others, in people who have succeeded in reducing their body weight. In-depth interviews have indicated that symptoms of this anxiety can persist for a long time, even after successful weight-loss treatment. Anxiety also affects adults who are not obese but were overweight in their youth. Analysis of the issue and interviews among patients also revealed the other side of anxiety—the anxiety about weight loss. This anxiety no longer applied to a specific clinical group and occurred much less frequently. Therefore, the aim of the presented study was to develop a two-dimensional scale for the diagnosis of weight-related anxiety, to develop an English version of the scale and to preliminarily test the psychometric properties of the emergent constructs. It is our intention that the scale could be used as a screening tool to identify individuals who need psychological support at an earlier stage to prevent the development of an eating disorder.

## 2. Study 1—Scale Design

### 2.1. Procedure for Developing Scale Items

In constructing this tool, the typical procedure based on the development of core items by expert panels was abandoned. We intended to build a tool that accurately describes weight-related anxiety and can be understood by a broad spectrum of patients of different ages. Firstly, in-depth interviews were used with patients attending dietetic and psycho-dietetic clinics. During these interviews, patients were asked whether they felt any emotions about their body, its shape or its weight. The answers were recorded and grouped in a further step. Interviews were collected from 134 individuals. In addition, internet postings on forums, blogs, and online counselling services regarding weight-related anxiety were analyzed. The following phrases were used to search for posts: anxiety, mass, body shape, too much weight, too low weight. Most posts were obtained from the following forums: abczdrowie.pl; wizaz.pl; sfd.pl; netkobiety.pl; hltv.org; mentalhealthforum.net; forumhealth.com. Examples of the posts were as follows:


*“…, I have a problem, I recently finished a job (I was on the move) which made me lose a bit of weight by eating less, now I have been out of work for a few days (for now) and I am practically not active at all, just sometimes I clean up, go out with the dogs, I eat relatively little and I am very afraid of getting fat (I have a mental problem with this), if I limit my food will I not get fat? The thought of getting fat in a few weeks scares me.”*

*“… I can’t gain weight I am 174 cm tall and 55 kg how can I help myself? I avoid meeting people because they keep asking me why I’m so skinny. I hate it, I’m afraid I’ll never get fat …”*

*“I overeat and I can’t control it, when I try to limit food, I get the fear that I’m going to lose weight and yet I already weigh too much …”*


The collected patient reports detailing the areas and situations in which the described anxiety occurs were compiled into unified items forming scales. The sentences were reformulated in such a way that they did not lose their original meaning and the respondent was able to assess how much a given situation concerned him or her on a dedicated scale. Only sentences prepared in this way and their prototypes were evaluated by competent experts specializing in psycho-dietetics. The prepared material included 54 sentences (items) describing anxiety-provoking situations and ways in which anxiety manifests itself. It was also decided to include sentences that did not clearly indicate weight-related anxiety, but nevertheless appeared in the patients’ accounts. For example, “Trying on new clothes, I do not feel comfortable” or “I have trouble controlling my body anxiety”. The aim of this procedure was to avoid losing components that might have been important for the understanding and accurate processing of the constructs. The material prepared in this way was subjected to the standard procedure for testing psychometric tools.

### 2.2. Participants and Procedure

The study was conducted on a group of 77 subjects, including 45 women (58%) and 32 men (42%). The mean BMI of the study group was 23.9; SD = 3.6. The mean age of the individuals was 30.9; SD = 9.8 and ranged from 18 to 50 years. According to the BMI criterion, 50 participants had normal body weight, 20 participants had higher than normal body weight and 7 participants had BMI below normal range. Participants were recruited from the student bodies of several Polish universities. Participants were provided with information about the purpose of the study and the possibility to withdraw consent to the study at any time without consequences. After consenting to the study, participants were given a questionnaire containing the study scale. Individuals with diagnosed eating disorders or mental illnesses were excluded from the study on the basis of their own declaration.

### 2.3. Results and Discussion

In order to determine the psychometric scale properties and the number of factors, an initial exploratory factor analysis and a reliability analysis were conducted. Once the items constituting the scale had been extracted, a time-series analysis and confirmatory factor analysis (CFA) were conducted in a second step to confirm the results obtained in the first step. To determine whether the collected dataset was appropriate to perform a factorial analysis, the KMO sample adequacy test and Bartlett’s sphericity test were used. The KMO value was 0.86, indicating an adequate sample. The result for the Bartlett sphericity test showed statistical significance (χ^2^ = 1846.45, *p*< 0.001), which provided the basis for conducting a factorial analysis. The two-factor structure of the tool emerged from the conducted analysis. The content item analysis indicated that the emerged components corresponded to weight gain anxiety and weight loss anxiety. Items that had weak loads were mainly related to general anxiety or fear of unspecified body change. Their loads were too weak to be taken into account in the analysis or to be able to extract another factor. For the final version, 20 items were selected including 10 items with the highest load for each subscale. The two-factor structure accounted for 70.7% of the total variance. Factor 1, explaining 39.3% of the variance, consisted of ten items which were marked as anxiety about getting fat (AGF). Factor 2 comprised ten items explaining 31.4% of the variance which were described as anxiety about losing weight (ALW). To assess the internal consistency of the scale, Cronbach’s alpha coefficients were 0.964 and 0.935 for the subscales and 0.916 for all scales. The rotated factor solution and Cronbach’s alpha coefficient are presented in Table 1. In a second step, confirmatory factor analysis (CFA) was carried out. It confirmed a very good fit of the model (RMSEA = 0.05; GFI = 0.99). Thirty of the participants in the original study were asked after one month to complete the scale again, which already contained only the selected 20 items. Correlation analysis of the original results and those from after the one-month interval revealed a statistically significant high correlation (r = 0.81; *p* < 0.01), indicating stability over time of the results obtained with the tool. The scale was named the body mass anxiety scale (BMAS-20) with the prefix PL—corresponding to the language in which it was constructed. It contains 20 items forming affirmative sentences. The respondent’s task is to indicate on a scale of 1–7 to what extent the sentence applies to him or her. In order to determine the result for a given scale, the scores for the items making up the scale should be added up. The scale does not contain reversed questions. The psychometric properties presented warrant a statement that it is a fully validated psychometric tool. The PL-BMAS-20 scale worksheet is attached as Appendix A and the key is provided in Appendix C.

## 3. Study 2—Translation and Validation of EN-BMAS-20 English Version

### 3.1. Scale Translation Procedure

The translation procedure was prepared taking into account previous translations of analyzed scales and principles used in cross-cultural studies. To ensure adequate linguistic accuracy, as a first step, the original tools were translated from Polish into English by three researchers fluent in both Polish and English. The results of these three translations were analyzed and discrepancies were identified. The meanings of some terms were modified or changed to improve understandability and cultural appropriateness. In the next step, two other bilingual experts performed the back translation (from English to Polish). The back-translated scales were compared by the English language native speaker with the original versions to confirm the accuracy and relevance of the translation. Once minor discrepancies were agreed upon, the final version of the translation was checked by three experts in psychology. Prepared translation was used to carry out a pilot study to confirm the comprehensibility and accuracy of the sentences. The pilot study was carried out on a group of 10 volunteer nursing students. The students did not indicate that the presented sentences were incomprehensible or that they had problems answering any question. 

### 3.2. Participants and Procedure

The participants in the study were a group of English-speaking students who were taking part in the ERASMUS program in Poland. The study was carried out on 69 students from different faculties. English was the national language of 20 participants. The others were fluent in English. The age of the respondents ranged between 19 and 25 years, with males accounting for 30% (21 persons). The majority of the subjects had a normal body weight (according to the BMI criterion) except for two subjects with a BMI of 25 and 25.5. Participants were recruited from the student bodies of several Polish universities. They were given information about the purpose of the study and the possibility to withdraw written consent at any time without consequences. Individuals with diagnosed eating disorders or mental illnesses were excluded from the study. After agreeing to participate in the study, individuals were given a questionnaire containing the scale.

### 3.3. Results and Discussion

To validate the English-language version of the EN-BMAS-20 scale and to confirm its psychometric properties, an exploratory factor analysis and a confirmatory factor analysis were conducted. In order to determine whether the collected dataset was appropriate to perform a factorial analysis, the KMO sample adequacy test and Bartlett’s sphericity test were used. The KMO value was 0.85, indicating an adequate sample. The result for the Bartlett sphericity test showed statistical significance (χ^2^ = 1136.47, *p* < 0.001), which provided the basis for conducting a factorial analysis. Analysis of the English version confirmed the two-factor structure of the tool representing weight gain anxiety and weight loss anxiety. The two-factor structure accounted for 64.8% of the total variance. Factor 1—AGF—explaining 38.75% of the variance, and contained ten items. Factor 2—ALW—comprised ten items explaining 26.1% of the variance. To assess the internal consistency of the scale, Cronbach’s alpha coefficients ranging 0.90 and 0.95 for the subscales and 0.91 for all scales. The rotated factor solution and Cronbach’s alpha coefficient are presented in Table 2. In a second step, a confirmatory factor analysis (CFA) was conducted which also confirmed a good fit of the model (RMSEA = 0.06; GFI = 0.95). The presented psychometric properties of the English-language scale version indicate comparable psychometric properties to the original version. The obtained results allowed us to conclude that the EN-BMAS-20 scale is a tool with good psychometric parameters. The EN-BMAS-20 scale sheet is attached as Appendix B and the key is attached as Appendix C.

## 4. Study 3—Characteristics of Anxiety Related to Weight Change

In recent years, intensive research has resulted with regard to the significance of weight gain anxiety for the onset and maintenance of eating disorders [24,25]. The aim of this study was to indicate the components of anxiety as measured by the BMAS-20 scale and to compare the characteristics of the two constructs in order to better understand their origins and the functions they play. To adequately represent the characteristics of the construct, without targeting a specific type of disorder, it was decided to study them on a non-clinical group. Variables were selected for the study that in previous studies indicated associations with body-related-anxiety constructs or were related to body-image construction, dietary use and nutrition.

### 4.1. Participants and Procedure

The study was conducted on a non-clinical group of 306 subjects, including 232 women (76%) and 74 men (24%). The mean BMI of the study group was 23.24; SD = 3.7. The mean age of the subjects was 27.62; SD = 9.68 and ranged from 18 to 58 years. There were 208 participants with normal body weight according to the BMI criterion, 82 participants with body weight above normal and 16 participants with BMI below normal range. Individuals were recruited from the student bodies of several Polish universities. Information was provided about the purpose of the study and the possibility to withdraw written consent at any time without consequences. After agreeing to the study, participants were given a set of questionnaires to complete including the study metric. Participants with diagnosed eating disorders or mental illnesses were excluded from the study on the basis of the respondent’s declaration. Respondents were also asked when they last weighed themselves and when they were last on a diet.

### 4.2. Methods

#### 4.2.1. Body Mass Index (BMI)

BMI is a correlation coefficient containing a quotient of the weight and square of height for the assessment of the correctness of nutritional status. According to the WHO, in adults, overweight is diagnosed at a BMI of 25–29.9 kg/m^2^ and obesity at a BMI ≥ 30.0 kg/m^2^. At normal weight, BMI is in the range 18.5–24.99 kg/m^2^. Underweight is diagnosed when BMI is below 18.5 kg/m^2^. The BMI of the study participants was calculated based on the weight and height values declared in the surveys.

#### 4.2.2. Self-Weight Status Perception (SWSP)

Subjective body weight assessment was made by responding to the question “My body weight is...” Respondents answered the question by choosing one answer from the following: too low; correct, too high.

#### 4.2.3. Personality

Personality traits of the subjects were determined using Costa and McCrae’s NEO-FFI personality inventory. NEO-FFI, which is based on the five-factor model of personality, is frequently used in employee training and vocational counselling as well as in professional development and personality development [26].

#### 4.2.4. Perceived Stress Level

The subjects’ level of perceived stress was measured using the PSS-10 scale by Sheldon Cohen, Tom Kamarck, Robin Mermelstein in the Polish adaptation by Juczyński. The scale contains 10 questions on various subjective feelings related to personal problems and events, behaviors and ways of coping with them [27].

#### 4.2.5. Anxiety and Depression

A screening questionnaire, the Hospital Anxiety and Depression Scale (HADS) by Zigmond and Snaith, was used to measure anxiety and depression. It has become a popular tool for clinical practice and research. It contains two scales whose levels correspond to the level of general anxiety and depression [28].

#### 4.2.6. Impulsiveness

The SUPPS-P Impulsive Behavior Scale was used to measure impulsivity. This scale contains three subscales: (1) emotion-based rash action, (2) lack of conscientiousness and (3) sensation seeking. The Polish version of the scale was developed by Ryszard Poprawa [29].

#### 4.2.7. Sensitivity to Punishment and Reward

The Sensitivity to Punishment and Sensitivity to Reward Questionnaire (SPSRQ) incorporates two scales: sensitivity to punishment and sensitivity to reward. The Polish adaptation of the questionnaire by Wytykowska, Białaszek and Ostaszewski was used [30].

#### 4.2.8. Self-Control

A 50-item self-control scale questionnaire, called the NAS- 50 by E. Nęcka, was used to measure self-control. This questionnaire consists of five subscales, called initiative and persistence (IP), proactive control (PC), switching and flexibility (SF), inhibition and adjournment (IA), and goal maintenance (GM) [31].

#### 4.2.9. Body Image

The Body Esteem Scale by S. Franzoi and S. Shields in the Polish adaptation by M. Lipowska and M. Lipowski was used to measure body image. It consists of 35 test items in three subscales. (1) physical attractiveness (PA) for males or sexual attractiveness (SA) for females, (2) upper body strength (UBS) for males or weight concern (WC) for females and (3) physical condition (PC) for both males and females [32].

#### 4.2.10. Data Analysis Procedure

Statistical analyses were performed using JASP 0.16.3. The data collected were first analyzed to determine which data were eligible for parametric tests [33]. For data whose parameters indicated that parametric tests could be used, Student’s T tests and r-Pearson correlation analysis were used for group comparisons. For the remaining data, Kruskal–Wallis Test and Dunn’s post hoc comparisons and rho-Spearman analysis were used.

### 4.3. Results and Discussion

#### 4.3.1. Variables Distribution and Their Relationship with Gender

Figure 1 shows the distribution of the variables of the weight change anxiety components and mean values in the study sample by gender. Gender statistically significantly differentiated the mean scores on the AGF scale (*p* < 0.01), while no statistically significant differences were noted on the ALW scale. The mean AGF score in the female group was higher (M = 31.88; SD = 17.64) than in the male group (M = 23.30; SD = 13.09). For these comparisons, Cohen’s d was 0.52. The observed effect size was classed as medium [34]. Although no gender differences were found in the ALW scale, it is important to note the differences in the distribution of the ALW variables in relation to the AGF. As can be seen in Figure 1, the frequency distribution of the AGF variables was flat with the highest frequency of the minimum value. In contrast, for the ALW scale, the values decreased more sharply and higher values in this scale occurred less frequently than in the AGF scale. In addition, for the AGF, the scatter of results was similar for both sexes. For the ALW, despite the lack of differences in mean values, the range of achievement among men was much greater. Despite the lack of gender-related differences in the mean values, higher scoring individuals on the ALW scale could be expected in the male group. Higher values of weight gain anxiety among women have also been confirmed in other studies. In addition, the impact of stereotypes and negative attitudes associated with obesity as confirmed in other studies is also stronger in women [15,35,36]. Fear of weight loss is such an under-researched construct, that we were unable to find studies presenting its gender dependence.

#### 4.3.2. Association of BMI and Self Perception of Weight Status with Body Mass Anxiety

In the next step, the mean values of the weight gain anxiety were analyzed in relation to the correctness of one’s body weight as assessed by BMI and subjective body weight assessment. The analyses indicated that both BMI and SWSP significantly differentiated the subjects in terms of their level of weight gain anxiety. Statistically significant differences were shown for BMI groups below normal BMI and normal BMI (*p* < 0.01), as well as below normal BMI and above normal BMI (*p* < 0.01). Only for the comparison of normal BMI and above normal BMI groups was no statistically significant difference shown (*p* = 0.07). In contrast, the comparison of the groups in terms of SWSP showed significant differences in AGF levels for all three groups at *p* < 0.01. Graphs showing mean analyzed values are shown in Figure 2.

Similar results were obtained after the analysis of mean anxiety values before weight loss. In this construct, BMI and SWSP also differentiated the subjects in levels of anxiety about weight loss, but statistically significant differences were shown for the normal BMI and above normal BMI (*p* < 0.01), as well as the above normal BMI and below normal BMI (*p* < 0.01) groups. Only for the comparison of normal BMI and below normal BMI groups was no statistically significant difference shown (*p* = 0.19). Conversely, the comparison of groups in terms of SWSP showed significant differences in ALW for all three groups at the *p* < 0.01 level. Mean analyzed values are shown in Figure 3.

Next, comparative analyses were performed on the mean AGF and ALW values in the group having a normal BMI. The subjects differed in mean AGF values according to their own perception of body weight (*p* < 0.01). The highest level of anxiety (M = 49.54; SD = 14.56) was in the individuals assessing their body weight as too high. The next group assessing their body weight as normal had a significantly lower score (M = 27.01; SD = 14.59). The lowest AGF scale score was found in the group assessing their body weight as too low (M = 16.50; SD = 7.03). Differences in mean ALW values were statistically insignificant. Figure 4 presents graphs showing these relationships.

The obtained results confirm the accuracy of the AGF and ALW constructs. One’s own perception of body weight and, in particular, the assessment of its regularity, is associated with the level of anxiety. Anxiety about gaining weight is higher in groups with an above normal BMI and in groups of respondents assessing their body weight as too high. In contrast, anxiety about losing weight is lower in groups with a below normal BMI and valuing their body weight as too low. The normal weight group also shows this trend.

#### 4.3.3. Association of Weighing Frequency and Dietary Adherence with Body Mass Anxiety

It was also examined whether there were differences in mean AGF and ALW values depending on the subjects’ declaration of when they last weighed themselves and when they were on a diet. The results of the mean values are shown in Table 3. The highest AGF values were found in the group of individuals who weighed themselves the least frequently (more than a year ago). In contrast, the highest ALW values were found in subjects who weighed themselves within the last month. The differences in both AGF and ALW values were statistically insignificant.

The mean values of the AGF and ALW scores were also analyzed according to the respondent’s declaration of when he or she had last been on a weight loss diet. As expected, the results of the ALW scale showed no significant differences. On the AGF scale, the highest values were achieved by respondents who had weighed themselves within the last month (M = 39; SD = 16.60). Significantly different results were noted in the groups that had never been on a diet (*p* < 0.01) and had been on a diet within the last year (*p* = 0.01). Significant differences were still noted between the groups of individuals who had never been on a diet and those who had been on a diet but could not remember when it was (*p* = 0.04). The remaining differences were statistically insignificant. The results are shown in Table 4.

#### 4.3.4. Correlation Analysis of Body Mass Anxiety Scales with Other Variables

An analysis of AGF and ALW co-occurrence with other psychological variables, as well as age and BMI, was performed. The analysis results are shown in Table 5. Age did not correlate with any of the studied factors. AGF correlated with most of the studied variables, with the strongest correlations occurring with body image (weight concern and physical condition subscales) and general anxiety as measured by the HADS scale. However, the AGF scale correlated significantly at an average level with perceived stress (PSS-10); impulsivity (emotion-based rash action subscale); neuroticism, self-control (IA subscale) and body image (upper body strength subscale). There was also a correlation with the level of sensitivity to punishment. The presented correlations and the previously presented results indicate that the AGF scale is strongly associated with the perception of one’s own corporeality. In addition to its relationship with the body image construct, it also shows a relationship with the variables characteristic of people struggling with overweight and obesity such as low conscientiousness or high impulsivity. High impulsivity in obese individuals is thought to result from an interaction between high reward sensitivity and poor self-control [37,38]. In our study, we observed the above correlations with AGF. However, in contrast to the above correlation, we found a significant positive correlation of AGF with punishment sensitivity (r = 0.37; *p* < 0.01). The research indicates that sensitivity to punishment generally has a protective function [39]. An additional analysis of this correlation, taking into account BMI, was performed. This analysis showed that the positive correlation of AGF with punishment sensitivity was only significant in the normal BMI group (r = 0.46; *p* < 0.01). In contrast, a significant correlation of ALW with punishment sensitivity occurred in the group with below-normal BMI (r = 0.88; *p* < 0.01). We, therefore, concluded that in some individuals AGF may have precisely a protective function linked to awareness of the negative consequences of poor nutrition and the risks associated with obesity and/or the consequences of obesity in social terms. This is certainly an area that needs to be further explored and researched to determine the function of this relationship. On the ALW scale, there were fewer correlations—the strongest, at the average level, occurred positively with levels of depression, negatively with levels of extraversion, agreeableness, and self-control (subscale SF). Despite several other significant correlations, the strength of these associations was close to zero. The lack of correlation with the level of general anxiety as measured by the HADS scale was surprising. This may be due to a specific construct that perhaps should not be treated as an anxiety state. The data described in literature suggest a possible link between weight loss anxiety and health status or health stereotypes [40]. Findings from a study of South African women indicated that respondents preferred to be overweight and bear the risk of acquiring cardiovascular disease, rather than being thin and stigmatized as having HIV or AIDS. Consequently, there was an observed lack of motivation to engage in physical activity for fear of losing weight and being stigmatized for it [41,42]. It can also be suspected that both ALW and AGF may be related to the anxiety of losing self-control. This anxiety is indicated in some studies as the most significant predictor of eating disorder symptoms, not only in women with eating disorders [43]. The number of studies on weight loss anxiety is negligible, but as the analyses above show, it appears to be an important construct with an under-described and under-researched effect. This points to the need for a deeper examination of this area as one that may be important in the emergence of eating disorders.

## 5. Main Discussion

The most commonly discussed eating disorders linked to psychological factors are anorexia nervosa (AN) and bulimia nervosa (BN). Research indicates that AN is associated with personality factors and anxiety-depressive disorders [44]. Anxiety about weight gain is recognized in the diagnosis of AN, although it is no longer a prerequisite for diagnosis [20]. Patients with AN without fat phobia, compared to those experiencing anxiety, report lower levels of psychopathological symptoms, including depression, anxiety, and fewer bulimic symptoms and less food restrictiveness [25,45,46]. In contrast, BN is an eating disorder involving recurrent episodes of overeating followed by adverse compensatory behavior [47,48]. Studies have shown that anxiety about weight gain is associated with psychopathological behaviors [47]. Anxiety co-occurs with obesity [16]. Our study also confirmed this relationship. In addition, we found that not only did BMI differentiate the subjects in terms of AGF, but so did SWPS. In the groups with above-normal BMI, as well as in the groups assessing their body weight as too high, anxiety was significantly higher than in the other subjects.

The literature contains far less data on the anxiety about losing weight than about gaining weight. Possibly, this is a reflection of the problems associated with the increasing number of overweight people worldwide. The desire for low body weight is associated with social approval and thinness is seen as an expression of an ideal fit into modern beauty standards, of which the figure is also a marker. Thus, the desire for weight gain and deviation from the culturally established body aesthetic is difficult to imagine. However, it cannot be ignored that weight control can also take into account the aversion to a slim figure and generate a fear of losing weight. From the presented data, it can also be suggested that both ALW and AGF may be related to the anxiety about losing self-control.

Some current research on the Goldfarb Fear of Fat Scale indicates a better fit using a two-factor model including fear of gaining weight and fear of losing control over food/weight [23]. Anxiety about losing self-control is not without significance, indicated in some studies as the most significant predictor of eating disorder symptoms, and not only in women with eating disorders [43].

Much less attention is paid to stigmatization due to being too thin, even though thinness is as stable a characteristic as thickness [49,50]. However, it must be assumed that the consequences of underweight discrimination are equally distressing and painful. Underweight women reported similar psychosocial problems as overweight women [51]. Interestingly, stigmatization due to low body weight also affects men and influences their body image. For example, underweight men experienced dissatisfaction with their own appearance and apprehension in interpersonal situations similar to that of overweight women [52]. Lack of satisfaction with their own bodies was also observed among very thin young boys [53,54]. Care for the figure can therefore address both the anxiety about excessive weight and the anxiety about weight loss, and thus generate different patterns of behavior, including pathological behavior, related to the pursuit of a particular figure.

Stigmatization due to excessive body weight is certainly significant in the development of weight anxiety and has been widely described in the literature. Negative attitudes mainly affect women. Studies show that most of these women are not satisfied with their bodies. This is fostered by exposure to an idealized body image through the media, which reinforces weight-related concerns [55]. According to Odgen, around 70% of women have been on a diet at least once in their lives, and at any given time around 40% of them are on a diet [56]. Anxiety about weight gain is a reaction in response to stimuli including prejudice and stigmatization by the environment, the evaluation of which may be negatively associated with increasing body weight. This can result in maladaptive behaviors such as eating restrictions or dietary regimes as a consequence of the weight gain anxiety [12,13].

The negative consequences of trying to achieve a slim figure include mood disorders, such as depression and anxiety. Associations between obesity and anxiety, as well as depression have been shown in a number of studies [16,57,58,59,60]. Weight stigma can promote the development of depressive-anxiety disorders through negative body self-esteem or discomfort in social relationships [61]. Symptoms of depression and anxiety were associated with eating disorders in both men and women, with this association being stronger in women. Overeating was also characteristic for men, and compensatory behavior was more common in women with higher levels of anxiety [62]. Results from an 11-year longitudinal study indicated that the initial presence of anxiety or depression was a predictor of increased weight gain [63]. The meta-analyses also suggest that these variables, rather than gender or age, are key to the development of obesity [16,64].

It also appears that treatment for psychiatric disorders may favor weight gain in patients due to the negative effects of medication. However, research is inconclusive in determining whether the relationship between body weight and anxiety or depressive disorders is bidirectional and whether the presence of one factor increases the risk of developing the other [20,65,66]. In our study, we observed an association of both AGF and ALW with levels of depression. These correlations were some of the stronger ones that could be indicated in this study. In contrast, we observed positive correlations of AGF and ALW with punishment sensitivity in different BMI groups. Research indicates that sensitivity to punishment generally has a protective function [39]. We therefore conclude that, in some individuals, anxiety about both AGF and ALW may have precisely a protective function linked to awareness of the negative consequences of poor eating and the risks associated with obesity. Moreover, in the group with normal BMI, a significant correlation of AGF with sensitivity to punishment was found. As with any trait, only excessively high levels of anxiety can be a predictor of psychopathology. These correlations need to be further analyzed to determine whether this function actually occurs, how effective it is, and whether it is still conditioned by other factors.

Despite widespread exposure to messages promoting thinness as a socially desirable trait, not everyone internalizes this message. Research among Latinos indicates that the popularization of the ideal slim beauty is not necessarily accepted by communities outside of Western culture. Distancing oneself from the slim figure promoted by the media may be fostered by an ethnic identity that favors more generous shapes [67]. In Ghana, fuller figures were also favored and these set the standard for body beauty, with pressure from the immediate environment directed towards achieving greater body mass [68]. It should be noted that a focus on achieving a greater body size can promote overweight and obesity and be associated with a number of co-morbidities. Similarly, a dietary regime carries with it numerous negative health consequences. In both cases—the pursuit of thinness or a fuller figure—achieving and maintaining a healthy body weight can be difficult, and body dissatisfaction is associated with discomfort and inappropriate eating behaviors.

## 6. Conclusions

The BMAS-20 scale in Polish and English versions is a fully useful psychometric tool.

AGF and ALW may have a protective function linked to awareness of the negative consequences of poor nutrition and the risks associated with obesity. Only above-normal levels of anxiety can be a predictor of psychopathology.

Both AGF and ALW are associated with symptoms of depression.

## 7. Limitation

The main limitation of the present study is self-declared weight and height by the subjects. These data were used to calculate BMI. In our further studies, independent anthropometric measurements should be taken to eliminate this limitation.

## Figures and Tables

**Figure 1 ijerph-20-02888-f001:**
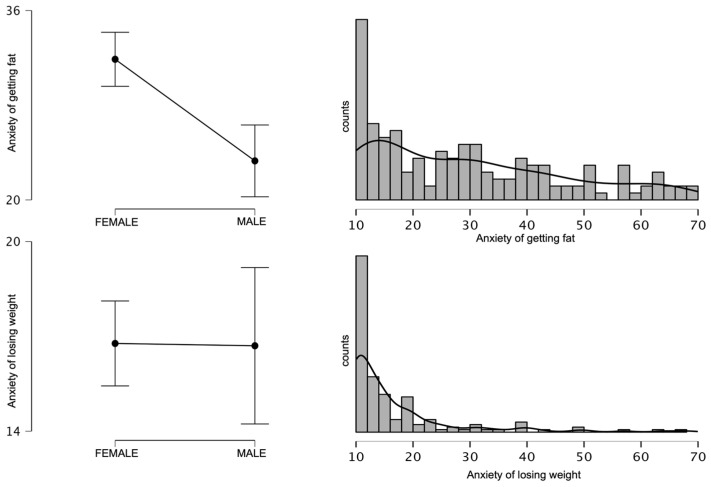
Gender differences and frequencies of the variables anxiety about getting fat and anxiety about losing weight.

**Figure 2 ijerph-20-02888-f002:**
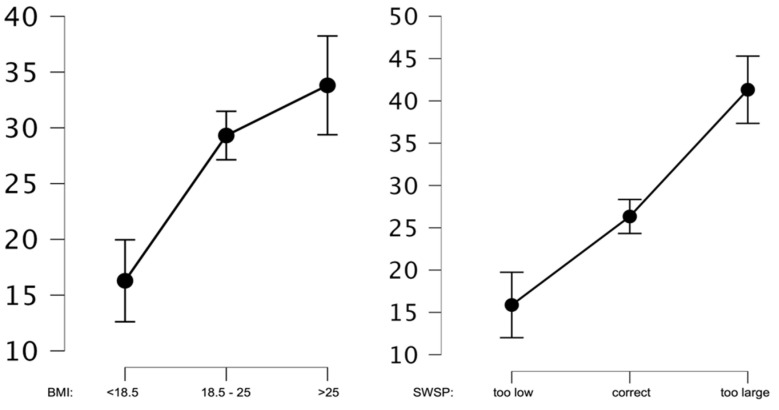
Average AGF values for BMI and SWSP.

**Figure 3 ijerph-20-02888-f003:**
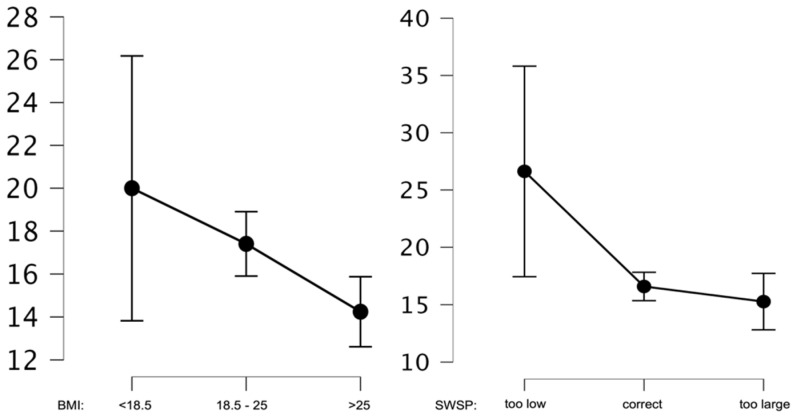
Average ALW values for BMI and SWSP.

**Figure 4 ijerph-20-02888-f004:**
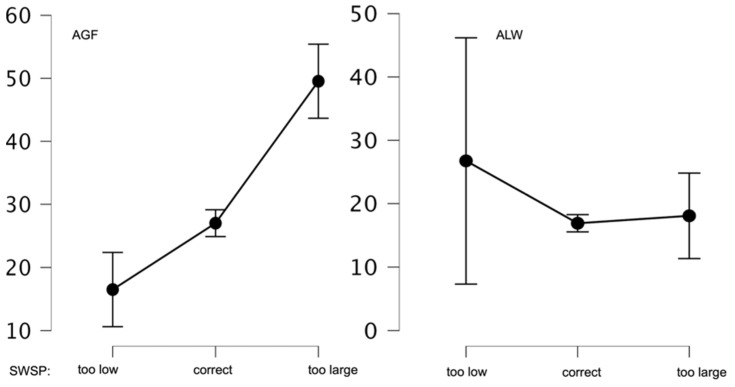
Mean AGF and ALF values of SWSP in the group with normal BMI.

**Table 1 ijerph-20-02888-t001:** Exploratory Factor Analysis of PL-BMAS-20.

Items	Anxiety about		Alfa Cr/% of the Variance	
	Getting Fat	Losing Weight		
7. Boje się czasami na siebie patrzeć z obawy, że przytyłam/łem. (Sometimes I am afraid to look at myself because I’m afraid that I have gained weight)	0.925		0.964/39.3%	0.916/70.7%
17. Nawet niewielki wzrost wagi powoduje u mnie lęk. (Generally, I avoid looking in the mirror so that it does not turn out that I have lost weight)	0.893			
8. Czasami mam wrażenie, że tyję od samego oddychania. (Sometimes it feels like I’m getting fat just from breathing)	0.885			
20. Nie lubię się ważyć, bo obawiam się, że przytyłam/em. (I do not like to weigh myself because I am afraid that I have gained weight)	0.878			
4. Po każdym nawet małym posiłku mam wrażenie, że przybyło mi ciała (After each, even a small meal, I have the feeling that I have gained weight)	0.865			
11. Są dni, kiedy patrząc na siebie np. w lustrze mam wrażenie, że znowu przytyłam/łem (There are days when looking at myself, e.g., in the mirror, I have a feeling that I have gained weight again)	0.858			
19. Boję się wejść na wagę, żeby się nie okazało, że się zmieniła (I’m afraid to step on the scale because it could turn out that my weight has changed)	0.855			
12. Po zjedzeniu zdarza mi się mieć wrażenie, że jestem od razu grubsza/y (After meals, sometimes I have the impression that I am immediately fatter)	0.851			
9. Jedząc czuję napięcie związane wpływem posiłku na wygląd mojego ciała (During meals I feel stress about how it will influence my body)	0.838			
1. Czasami jak wydaje mi się, że przytyłam/łem to czuję negatywne napięcie i ogarniający mnie smutek (Sometimes when it seems to me that I have gained weight, I feel negative tension and overwhelming sadness)	0.834			
15. Zdarza się, że jem na siłę, żeby tylko nie schudnąć (It happens that I force myself to eat to not lose the weight)		0.925	0.935/31.4%	
16. Unikam ważenia się, bo wywołuje u mnie lęk, że schudłam/em (I avoid weighing myself because it makes me afraid that I have lost weight)		0.882		
5. Gdy widzę, że moja masa ciała trochę spadła czuję lęk (When I see that my body weight has dropped a little, I feel anxious)		0.878		
6. Czuje, że moje ciało jest jak szkielet i nie jest atrakcyjne (I have the feeling that my body is like a skeleton and is not attractive)		0.850		
17. Ogólnie unikam oglądani się w lustrze, żeby nie okazało się że schudłam/em (Generally, I avoid looking in the mirror so that it does not turn out that I have lost weight)		0.832		
14. Obawiam się sytuacji jedzenia z innymi i ich komentarzy, że nic nie jem i jestem chuda/y (I’m afraid of having meals with others and their comments that I don’t eat anything and I’m skinny)		0.772		
2. Boję się, że w przyszłości schudnę i nic na to nie poradzę (I am afraid that in the future I will lose weight and I cannot do anything about it)		0.753		
13. Bywają dni, kiedy patrząc na swoje ciało w lustrze mam wrażenie, że znowu schudłam/łem (There are days when looking at my body in the mirror I have the impression that I lost weight again)		0.739		
3. Czasami jak wydaje mi się, że schudłam/łem czuję, że tracę radość życia (Sometimes when it seems to me that I have lost weight, I feel that I am losing the joy of life)		0.727		
10. Przejadam się i nie umiem tego kontrolować, inaczej pojawia się lęk że stracę kilogramy (I overeat and I can’t control it, otherwise I’m afraid that I will lose weight)		0.666		

**Table 2 ijerph-20-02888-t002:** Exploratory factor analysis of EN-BMAS-20.

Items	Anxiety about		Alfa Cr/% of the Variance	
	Getting Fat	Losing Weight		
20. I do not like to weigh myself because I am afraid that I have gained weight.	0.921		38.75%/0.949	64.88%/0.908
18. Even a slight increase in weight makes me anxious.	0.920			
12. After meals, sometimes I have the impression that I am immediately fatter.	0.918			
7. Sometimes I am afraid to look at myself because I’m afraid that I have gained weight.	0.872			
11. There are days when looking at myself, e.g., in the mirror, I have a feeling that I have gained weight again.	0.846			
9. During meals I feel stress about how it will influence my body.	0.810			
4. After each meal, even a small one, I have the feeling that I have gained weight.	0.797			
19. I’m afraid to step on the scale because it could turn out that my weight has changed.	0.748			
1. Sometimes when it seems to me that I have gained weight, I feel negative tension and overwhelming sadness.	0.737			
8. Sometimes it feels like I’m getting fat just from breathing.	0.700			
3. Sometimes when it seems to me that I have lost weight, I feel that I am losing the joy of life.		0.900	26.13%/0.900	
16. I avoid weighing myself because it makes me afraid that I have lost weight.		0.837		
5. When I see that my body weight has dropped a little, I feel anxious.		0.833		
6. I have the feeling that my body is like a skeleton and is not attractive.		0.767		
17. Generally, I avoid looking in the mirror so that it does not turn out that I have lost weight.		0.738		
15. It happens that I force myself to eat to not lose the weight.		0.727		
14. I’m afraid of having meals with others and their comments that I don’t eat anything and I’m skinny.		0.720		
13. There are days when looking at my body in the mirror I have the impression that I lost weight again.		0.631		
2. I am afraid that in the future I will lose weight and I cannot do anything about it.		0.630		
10. I overeat and I can’t control it, otherwise I’m afraid that I will lose weight.		0.546		

**Table 3 ijerph-20-02888-t003:** Frequency of weighing and mean values of AGF and ALW.

	AGF		ALW	
When Was the Last Time You Weighed Yourself?	M	SD	M	SD
More than year ago	34.00	12.63	12.00	1.69
During the last six months	24.46	14.20	16.46	8.99
In the last two weeks	28.53	16.24	15.87	7.02
In the last month	32.00	16.56	20.64	15.8
In the last week	29.22	16.27	17.43	10.2

**Table 4 ijerph-20-02888-t004:** Frequency of dietary intake and mean AGF and ALW values.

	AGF		ALW	
Have You Ever Been on a Diet for Weight Loss?	M	SD	M	SD
No	24.64	15.11	18.61	14.17
Yes, in the last month	39.00	16.60	15.57	6.47
Yes, in the last six months	35.00	17.40	14.00	7.56
Yes, over the past year	17.00	3.58	13.67	4.93
Yes, but I don’t remember when	31.33	13.04	19.00	9.54

**Table 5 ijerph-20-02888-t005:** Correlation of BMAS scale factors with other factors.

	AGF	ALW
AGE	0.00	0.09
BMI	0.25 ***	−0.03
PSS-10	0.41 ***	0.11
HADS_Depression	0.26 **	0.24 **
HADS_Anxiety	0.50 ***	0.12
SUPPS_ERA	0.44 ***	0.07
SUPPS_LC	0.27 **	0.03
SUPPS_SS	−0.06	−0.08
SPSRQ_P	0.37 ***	0.06
SPSRQ_R	0.11	0.06
NEO FFI_N	0.42 ***	0.05
NEO FFI_E	−0.05	−0.25 ***
NEO FFI_O	0.11	−0.07 **
NEO FFI_A	0.05	−0.20 *
NEO FFI_C	−0.22 **	−0.12 *
NAS-50_IP	−0.16	0.03
NAS-50_PC	0.00	−0.11
NAS-50_SF	−0.16	−0.26 ***
NAS-50_IA	−0.40 ***	0.04 **
NAS-50_GM	−0.20 *	0.02
BES_PA	−0.35 ***	−0.11
BES_UBS	−0.46 ***	−0.24
BES_PC	−0.54 ***	0.04 ***
BES_SA	−0.34 ***	−0.14
BES_WC	−0.60 ***	0.06 ***

* *p* = 0.05; ** *p* = 0.01; *** *p* < 0.01.

## Data Availability

Data is contained within the article.

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
