# Peer review of "I Don’t Want to Be Thin! Fear of Weight Change Is Not Just a Fear of Obesity: Research on the Body Mass Anxiety Scale"

_ijerph, 2023, doi:10.3390/ijerph20042888_

Round 1

Reviewer 1 Report

The paper makes a very interesting read and clearly fills in a missing gap in the research field. It is evident that there needs to be more research conducted on the fear of losing weight; the current paper addresses the issue well and provides a great starting point for subsequent research. 

The authors clearly state that there is no research being done on the topic; however, a current search on any new studies might be useful in order to be fully sure of this. If there are any studies, it would be interesting to see what they found out. 

Author Response

Thank you for the good feedback on our work. As we indicated, fear of weight gain is a well-known construct. In contrast, fear of weight loss is poorly studied. Since the manuscript was sent, no more research has been published on this topic. We consider our work as preliminary research, mainly oriented towards the conversion of tools to study the indicated variables. We hope that the published tools will allow us to fully investigate the phenomena presented. 

Reviewer 2 Report

The review comments are listed as follows.

1.        The 54 sentences (items) describe anxiety-provoking situations and ways that anxiety manifests itself in Subsection 2.1, the authors should have more descriptions to explain how they got those sentences.

2.        Some abbreviations should have corresponding full texts when they appear in the manuscript first.

3.        The authors should have a legend to explain the meaning of the symbol ‘-‘.

4.        Line 227, “conducted.In” should be “conducted. In”.

5.        Examining Table 2, the two-factor structure of the total variance should be 64.88%, instead of 64.8 %.

6.        In the study related to characteristics of anxiety related to weight change, the authors should explain the reasons why they use those methods (BMI, SWSP, Personality, Perceived stress level, Anxiety and Depression, Impulsiveness, Sensitivity to Punishment and Reward, Self-control, and Body Image).

7.        What are the units of the X-axis and the Y-axis of the figures of AGF and ALW shown in Figure 1?

8.        Figure 3 shows the average ALW values for BMI and SWSP; why do not the authors show the average AGF values for BMI and SWSP?

9.        It had better for the authors to have a legend in Figures 1 ~ 4 to explain the meanings of the graphic symbols in those figures.

10.    Please confirm the data mentioned in lines 386 – 390 is corresponding to Figure 4.

11.    The authors should describe how they get the data shown in Table 3 and Table 4.

12.    Some data in Table 5 are overlapped. The authors should fix this mistake.

13.    In general, the content of the conclusion section should be expressed in a narrative way rather than a list way.

14.    The authors might ask a professional English editor to revise their manuscript.

Author Response

  1. The 54 sentences (items) describe anxiety-provoking situations and ways that anxiety manifests itself in Subsection 2.1, the authors should have more descriptions to explain how they got those sentences.

The whole 2.1 subsection describes how the items are constructed. In addition, we have expanded the description of obtaining data. We hope that the expanded description will be satisfactory to the reviewer.

  1. Some abbreviations should have corresponding full texts when they appear in the manuscript first.

Abbreviations appear for the first time in chapter 2.3 and have the full development and the abbreviation in brackets. For a better reading of the paper, we have additionally included an abbreviations section at the end of the manuscript.

  1. The authors should have a legend to explain the meaning of the symbol ‘-‘.

Unfortunately, we do not know which symbol is meant - if it is the '-' in the scale name, the way the name is constructed is described in sections 2.3 and 3.3. In addition, we have included an explanation of the abbreviation of scale names in the abbreviations section. This symbol between abbreviations or names should be treated as a dash.

  1. Line 227, “conducted.In” should be “conducted. In”.

Suggested change has been made.

  1. Examining Table 2, the two-factor structure of the total variance should be 64.88%, instead of 64.8 %.

In the manuscript, in Table 1 total variance is entered as 64.88%. Perhaps there was a problem displaying the pdf. In MSWorld everything is correct.

  1. In the study related to characteristics of anxiety related to weight change, the authors should explain the reasons why they use those methods (BMI, SWSP, Personality, Perceived stress level, Anxiety and Depression, Impulsiveness, Sensitivity to Punishment and Reward, Self-control, and Body Image).

Selected variables in the presented study were chosen taking into account previous studies indicated associations with the body-related anxiety constructs under investigation or were related to body image construction, dietary use and nutrition. The description was completed.

  1. What are the units of the X-axis and the Y-axis of the figures of AGF and ALW shown in Figure 1?

The descriptions have been completed.

  1. Figure 3 shows the average ALW values for BMI and SWSP; why do not the authors show the average AGF values for BMI and SWSP?

­­­ The average values for the AGF are presented in Figure 2.

  1. It had better for the authors to have a legend in Figures 1 ~ 4 to explain the meanings of the graphic symbols in those figures.

We do not quite know which graphic symbols are meant. The abbreviations used are introduced much earlier in the text and in addition we have introduced an explanation of the abbreviations at the end of the manuscript.

  1. Please confirm the data mentioned in lines 386 – 390 is corresponding to Figure 4.

Yes, we confirm it.

  1. The authors should describe how they get the data shown in Table 3 and Table 4.

The description in 4.1 and 4.3.3 has been completed.

  1. Some data in Table 5 are overlapped. The authors should fix this mistake.

We do not know what overlapping data is involved - when analyzing the data we do not see the overlap.

  1. In general, the content of the conclusion section should be expressed in a narrative way rather than a list way.

In our experience, the letter method is more readable. The drafting guidelines do not require the conclusions to be presented in a different way than we have done.

  1. The authors might ask a professional English editor to revise their manuscript.

The manuscript has been proofread by a native speaker.

Reviewer 3 Report

This is an interesting and methodologically sound paper on ever important topic. It is indeed important to have an approprate measure to body-related anxiety.

The paper needs minor proofreading: for instance, the lack of full stops in the abstract.

Abstract: anxiety can be associated with body weight but ut is not the only psychological factor. Please rephrase the first phrase. 

Please put the keywords in alphabetic order.

Introduction: I am not sure that it is a uniquely western issue.

Line 87: an interview? just one?

Study 1: the qualitative part is very vague: how many interviews, what blogs? Please elaborate. 

Study 3: the variables operationalized in study 3 and their interrelationships should be better explained. It would be more interesting to test the predictive value of the new scale vs BMI according to the same model. 

Best of luck with your research! 

Author Response

This is an interesting and methodologically sound paper on ever important topic. It is indeed important to have an approprate measure to body-related anxiety.

Thank you for this opinion, in our opinion the topic is also very interesting.

The paper needs minor proofreading: for instance, the lack of full stops in the abstract.

Abstract: anxiety can be associated with body weight but ut is not the only psychological factor. Please rephrase the first phrase.

Changes have been made.

Please put the keywords in alphabetic order.

Changes have been made.

Introduction: I am not sure that it is a uniquely western issue.

The sentence has been deleted it adds nothing to the construct under examination

Line 87: an interview? just one?

Not just one, of course – it was corrected.

An in-depth interviews indicated that symptoms of this anxiety can persist for a long time, even after successful weight loss treatment.

Study 1: the qualitative part is very vague: how many interviews, what blogs? Please elaborate.

The description in the manuscript has been completed.

Study 3: the variables operationalized in study 3 and their interrelationships should be better explained. It would be more interesting to test the predictive value of the new scale vs BMI according to the same model.

The main aim of the article was to present tools for the study of the variables presented. The research describing the characteristics of the anxiety construct is preliminary. We realize that in order to fully describe this construct, it is important to extend the research, which we will do in the next steps. Thank you for your suggestions - we will certainly use them in our future research and publications.